# Disintegration of the NuRD Complex in Primary Human Muscle Stem Cells in Critical Illness Myopathy

**DOI:** 10.3390/ijms24032772

**Published:** 2023-02-01

**Authors:** Joanna Schneider, Devakumar Sundaravinayagam, Alexander Blume, Andreas Marg, Stefanie Grunwald, Eric Metzler, Helena Escobar, Stefanie Müthel, Haicui Wang, Tobias Wollersheim, Steffen Weber-Carstens, Altuna Akalin, Michela Di Virgilio, Baris Tursun, Simone Spuler

**Affiliations:** 1Muscle Research Unit, Experimental and Clinical Research Center, A Joint Cooperation of the Charité Universitätsmedizin Berlin and the Max Delbrück Center for Molecular Medicine in the Helmholtz Society, Lindenberger Weg 80, 13125 Berlin, Germany; 2Charité Universitätsmedizin Berlin, Department of Pediatric Neurology, 13353 Berlin, Germany; 3Berlin Institute of Health–Universitätsmedizin Berlin, 10117 Berlin, Germany; 4Laboratory of DNA Repair and Maintenance of Genome Stability, Max Delbrück Center for Molecular Medicine in the Helmholtz Association, 13092 Berlin, Germany; 5Berlin Institute of Medical Systems Biology (BIMSB), Max Delbruck Center for Molecular Medicine in the Helmholtz Association, 10115 Berlin, Germany; 6Charité Universitätsmedizin Berlin, Department of Anesthesiology and Operative Intensive Care Medicine, 13353 Berlin, Germany

**Keywords:** epigenetic, muscle stem cell, critical illness, histone 1

## Abstract

Critical illness myopathy (CIM) is an acquired, devastating, multifactorial muscle-wasting disease with incomplete recovery. The impact on hospital costs and permanent loss of quality of life is enormous. Incomplete recovery might imply that the function of muscle stem cells (MuSC) is impaired. We tested whether epigenetic alterations could be in part responsible. We characterized human muscle stem cells (MuSC) isolated from early CIM and analyzed epigenetic alterations (CIM *n* = 15, controls *n* = 21) by RNA-Seq, immunofluorescence, analysis of DNA repair, and ATAC-Seq. CIM-MuSC were transplanted into immunodeficient NOG mice to assess their regenerative potential. CIM-MuSC exhibited significant growth deficits, reduced ability to differentiate into myotubes, and impaired DNA repair. The chromatin structure was damaged, as characterized by alterations in mRNA of histone 1, depletion or dislocation of core proteins of nucleosome remodeling and deacetylase complex, and loosening of multiple nucleosome-spanning sites. Functionally, CIM-MuSC had a defect in building new muscle fibers. Further, MuSC obtained from the electrically stimulated muscle of CIM patients was very similar to control MuSC, indicating the impact of muscle contraction in the onset of CIM. CIM not only affects working skeletal muscle but has a lasting and severe epigenetic impact on MuSC.

## 1. Introduction

Admission to intensive care units (ICU) is often complicated by critical illness myopathy (CIM), characterized by progressive weakness and permanent myopathy [1,2]. CIM prolongs the time of mechanical ventilation, facilitates hospital comorbidities like pneumonia, adds significantly to the length of the hospital stay, and increases the risk of death [3]. Patients commonly fail to recover despite professional rehabilitation programs. Muscle weakness remains persistent, and the patients become chronically disabled [4]. The name of the disease entity of high medical need is not consistently used. ICU-acquired weakness (ICUAW) and CIM describe the disorder. CIM is appropriate if electrophysiological studies were performed to identify the problem while the patient was still unconscious. ICUAW requires an objective evaluation of muscle strength. 

After 5 years, only 44% of ICU survivors have returned to work, demonstrating the negative economic and quality-of-life impacts [5]. Why muscle regeneration fails to recover is not understood. However, muscle stem cells, also termed satellite cells, may be involved [6]. Many mechanisms contribute to CIM, including immobilization, disturbed glucose homeostasis, inflammation, inflammatory cytokines, and muscle-fiber myosin breakdown triggered by various pathways [6,7,8,9,10]. However, toxic influences on sarcomere structures, stress, and metabolic alterations are no longer relevant once a patient has left the ICU. Therefore, we asked whether or not a more permanent damage mechanism could be responsible for the refractory regenerative capacity of skeletal muscle during critical illness that might prevent successful and effective rehabilitation. 

In general, skeletal muscle exhibits a high regenerative capacity. Although the loss of muscle mass accompanies aging to some degree, regeneration is nonetheless possible [11]. Skeletal muscle harbors its own stem-cell population. These cells are located in a niche between the sarcolemma and basal lamina of skeletal muscle fibers. Satellite cells are sparse and make up only 2–4% of skeletal muscle nuclei. However, they are powerful and indispensable for muscle regeneration. Satellite cells are characterized by the transcription factor PAX7 [12,13,14,15,16]. Once activated, satellite cells proliferate, differentiate while expressing further myogenic markers like MYF5 and MYOD, and fuse with existing muscle fibers to ensure the regenerative process [17,18,19,20].

Heterochromatin is the term applied to firmly packed, genetically inactive forms of DNA, while euchromatin is uncoiled, loosely packed, and transcriptionally active. Regulation of these two chromatin states is very important for maintaining cell specialization and function, a process of chromatin remodeling also known as “epigenetics” since these genetic alterations do not involve changes in the DNA sequence. One multiprotein complex of central importance in maintaining the exact level of transcriptional activity is called the nucleosome remodeling and deacetylase (NuRD) complex (Figure 1A). NuRD consists of 6–8 subunits whose exact composition is tissue specific. The main components of NuRD in muscle tissue are ATPase (chromodomain helicase DNA-binding protein 2 and 4; CHD2/4), the histone deacetylase HDAC1/2 (class I lysine deacetylase1/2), two metastasis tumor-antigen proteins (MTA), and the core component histone chaperones RB (retinoblastoma binding protein 4 and 7; RBBP4, RBBP7). In muscle, alterations in NuRD have been described as hereditary mutations in the nuclear envelope gene LMNA encoding lamin A/C [21].

The chromatin structure can also be affected by linker histone 1, which plays a pivotal role as an epigenetic regulator in establishing the compact state of the nucleosome assembly. It is known that alterations in histone 1 expression have been associated with the regulation of chromatin structure during the cell cycle, as well as transcriptional regulation, DNA damage response, and cellular differentiation [22,23,24].

We tested the hypothesis that epigenetic mechanisms could contribute to the failure of skeletal muscle regeneration in CIM patients. Our investigation focuses on muscle stem cells with high regenerative potential rather than post-mitotic muscle fibers. We investigated the function of skeletal muscle stem cells obtained from CIM patients during the first week of critical illness. Although the cells were nurtured under optimal laboratory conditions, we detected profound epigenetic alterations and functional changes in this pivotal cell population long after the muscle stem cells had been isolated from their potentially toxic, critically ill environment. MuSC obtained from skeletal muscle that had undergone electrical stimulation and therefore was less immobilized displayed an almost normal profile. These findings could also be relevant for other acquired muscle-wasting diseases like cachexia accompanying cancer, cardiomyopathy, chronic obstructive pulmonary disease, and space flight muscle atrophy.

## 2. Results

### 2.1. Impaired Growth and Differentiation of MuSC in Early CIM

We isolated MuSC from muscle biopsy specimens obtained early within the first week after ICU admission (Figure 1B; CIM *n* = 12; controls *n* = 18; patient information: Table 1 and Table 2 and Appendix A) using established protocols [11,25]. Analyzed cell populations were highly myogenic, as indicated by desmin expression of >98%. Morphologically, CIM-MuSC could not be distinguished from controls. However, more thorough analyses revealed major differences: First, the number of MuSC expressing the proliferation marker Ki-67 was significantly lower in CIM-MuSC (*p* < 0.001), thus indicating reduced proliferative capability. In agreement with this result, we confirmed impaired cell cycle progression by the carboxyfluorescein succinimidyl ester (CFSE) dilution assay (*p* < 0.01, Figure 1C). Second, the number of MuSC expressing the transcription factor PAX7 was highly reduced in CIM as compared to controls (*p* < 0.01, Figure 1D). Third, the ability of MuSC to differentiate into myotubes was significantly reduced in CIM-MuSC (in standard and serum depleted medium, each with *p* < 0.001) as determined by calculation of the fusion index in the absence of serum depleted medium (Figure 1E).

### 2.2. MuSC Obtained from Early CIM Lose Heterochromatin

After having established that CIM-MuSC had significant functional deficits, we investigated the epigenetic profiles of MuSC from CIM patients. Accordingly, we performed unbiased RNA-sequencing analysis, ATAC-Seq, confirmatory RT-PCR analysis, and Western blots on CIM-MuSC and controls. Major differences between CIM-MuSC and control MuSC were found with heat-shock-associated and immunologic genes, which were upregulated, as well as histone genes, which were downregulated (Appendix A). Strikingly, among the genes most significantly downregulated were 11 genes encoding histone 1, such as H1H3D, H1H3C, and H1H2AE. We confirmed these findings by qPCR (*p* < 0.05) and on protein level (Figure 2A). Histone 1 molecules are chromatin stabilizers and are responsible for heterochromatin formation. Since histone 1 depletion hints at epigenetic alterations in CIM-MuSC, we investigated whether the epigenetic status of other histones and their modifications was also altered in CIM-MuSC. An important marker associated with heterochromatin formation is the tri-methylation of lysin 9 on histone H3 (H3K9me3). CIM-MuSC had significantly fewer H3K9me3-positive myonuclei than controls (*p* < 0.001, Figure 2B). We then assessed whether downregulated histone genes (Table 3) and proteins were a global phenomenon in CIM-MuSC. Indeed, ATAC-Seq experiments demonstrated the loosening of multiple nucleosome-spanning sites in CIM-MuSC, as indicated by the downregulation of grouped fragment sizes at coding regions and the intragenic fragment length distribution (Figure 2C). This finding indicates a global loss of heterochromatin in CIM-MuSC or a general loosening of the densely packed chromatin structure, rendering genes accessible to transcription that would normally be inactivated. 

### 2.3. MuSC Accumulate DNA Double-Strand Breaks in CIM

Decondensed euchromatin domains are more susceptible to DNA damage than heterochromatin. We investigated whether CIM-MuSC genomes contained more DNA double-stranded breaks (DSB). Since gamma-phosphorylated H2AX (γH2AX) accumulates in large chromatin domains surrounding DNA breaks and is widely employed as a marker for DSB, we assessed DSB formation in CIM-MuSC by quantifying the presence of γH2AX foci. We detected a significantly higher number of γH2AX foci in CIM-MuSC compared to controls (*p* < 0.01). These data were also confirmed by Western blot analysis of γH2AX global protein levels (Figure 3A,B). DSB can be caused by increased susceptibility of the DNA to damage or by a reduced capability to repair DSB. We monitored the kinetics of the repair of irradiation-induced (IR) γH2AX foci in CIM-MuSC and controls. 24 h post-irradiation, control MuSC had successfully repaired close to 80% of IR-induced DSB, whereas, in CIM-MuSC, only 40% were repaired (*p* < 0.05), thus indicating a disturbed ability to repair DSBs (Figure 3C). Altogether, the data suggest that DSB accumulation may be an important factor in the dysfunction of CIM-MuSC. 

### 2.4. NuRD Subunits in MuSC Are Altered in CIM

Next, we asked which factors known to influence chromatin structure might be involved in CIM chromatin deconvolution. The NuRD complex is known to promote heterochromatin formation, and dysfunction of NuRD leads to transcriptional changes. We measured the NuRD complex members RBBP4 and MTA2. In contrast to controls, RBBP4 and MTA2 were depleted in a significant number of CIM-MuSC myonuclei (*p* < 0.001). Furthermore, MTA2, but not RBBP4, aggregated in the cytoplasm of CIM-MuSC. This corresponded with the quantification of protein levels by Western blot, showing downregulated RBBP4 expression (Figure 3D). Total MTA2 was expressed at similar levels implying that the faulty localization to the cytoplasm indeed took place and that the cytoplasmatic foci were not unspecific aggregates. It was previously shown that the knockdown of MTA2 leads to an accumulation of DSB and increased IR sensitivity [26]. Indeed, we observed MTA2 abutting γH2AX foci (Figure 3E). To confirm that the changes in CIM-MuSC are specific to CIM, we included MuSC obtained from three ICU controls who had not developed CIM (ICU controls). Distribution patterns of Ki-67, PAX7, γH2AX, H3K9me3, RBBP4, and MTA2 in immunofluorescence staining of MuSC from ICU controls resembled that of the healthy controls (Appendix A). We also asked whether or not CIM-MuSC underwent premature aging and compared CIM-MuSC to high-passage, aged, or senescent human primary MuSC (Appendix A). The morphology of senescent cells was different with larger cell size and vacuoles, but the DSB are more prevalent in CIM-MuSC. 

### 2.5. Reduced Muscle Regenerative Potential of CIM-MuSC

Epigenetic alterations and reduced in vitro differentiation capacity of CIM-MuSC should lead to an impairment to build new muscle fibers in vivo if our observations are of physiological relevance. Therefore, we transplanted CIM-MuSC into the irradiated anterior tibialis muscle of immunodeficient NOG mice. Three weeks after transplantation, we found that human nuclei that had not integrated into muscle fibers were significantly higher (36%) in CIM-MuSC than in transplants from control-MuSC (9%), implying that CIM-MuSC is less effective in creating the muscle fibers (Figure 4A,B). We also found that both CIM- and control MuSC were able to repopulate the stem cell niche (Figure 4C). 

### 2.6. MuSC Benefit from EMS in CIM

Next, we asked to which extent muscle contraction affects the epigenetic changes in MuSC. MuSC populations were isolated from quadriceps muscle that had undergone electrical muscle stimulation (EMS) [9], whereas the autologous cell population derived from the contralateral vastus lateralis had not. Whereas MuSC from the EMS muscle was almost identical to normal controls, MuSC from unstimulated muscle again showed the entire plethora of pathological alterations. We supported this conclusion with additional assays, i.e., proliferation capacity, the occurrence of DSB and NuRD complex distribution, and the ability to repair the DSB after irradiation (*p* < 0.001) (Figure 5).

Data information: Scale bar for all images: 50 µm. In (C,D), all values are expressed as mean ± SEM. In (C,D), data were analyzed by Mann–Whitney U test. At least 200 nuclei/cell population were counted for quantification. In (E), the data were analyzed using Chi-square test, at least 400 nuclei/cell population were counted for quantification. ** *p* < 0.01, *** *p* < 0.001.

## 3. Discussion

We show that within days after the onset of severe critical illness, MuSC exhibit substantial epigenetic alterations leading to severe functional impairment and ultimately to poor recovery from CIM. The known beneficial effect of muscle contraction has been confirmed and receives some explanations on a molecular mechanistic level [8,9,27]. Epigenetic alterations may have a permanent impact on gene expression and cell function. In the context of sepsis and acute lung injury, epigenetic alterations have been recognized based on the methylation profile of DNA on primary human monocytes and human airway epithelial cells [28,29]. It was demonstrated that the expression of more than 3700 genes was altered as soon as two hours after exposure to endotoxemia [30]. This observation was verified in subsequent studies that showed changes in DNA methylation and histone marks [28,29]. The dysregulated genes were involved in cytokine responses and interferon signaling. Thus far, only a few studies have addressed DNA methylation in murine MuSC and have shown that aging affects epigenetic drift or uncoordinated accumulation of methylation changes in promoter regions, resulting in increased transcriptional variability that impairs the ability of MuSC to maintain quiescence [31,32]. It remains unclear how stem cell molecular variability affects their proliferation and/or differentiation. 

Future studies of the DNA methylation status of muscle gene-related genomic regions in MuSC from CIM patients may lead to a better understanding of gene regulation in skeletal muscle in the context of the disease. In our study, the vast majority of MuSC populations were isolated from male donors. There is evidence that gender aspects influence the clinical outcome of ICU-acquired weakness [7,33]. Although there is no experimental evidence for gender aspects in muscle stem cell function, we cannot completely rule out a bias in our study. A role for histone modification was shown in a mouse model of sepsis, and treatment with histone deacetylase inhibitors appeared beneficial in improving lung injury, survival, and the level of circulating Il-6 in the blood [32].

The nucleosome remodeling and deacetylase complex (NuRD) is a multiprotein complex and a key modulator of chromatin and histone acetylation. It consists of six subunits with individual compositions in different tissues [21,33]. The interactions within the subunits of the NuRD are not well understood. The loss of some NuRD components leads to changes in the chromatin structure that mimic aging [24,34]. One might argue that the alterations we identified in CIM-MuSC were a premature aging phenomenon. However, morphological hallmarks of aging MuSC, such as enlargement and vacuolization, were not present in CIM-MuSC. The NuRD complex also plays a role in the DNA damage response. The genome is constantly exposed to various agents that induce DNA damage. Efficient repair of DNA damage is mandatory for cell survival. The NuRD complex promotes access to the sites of DNA damage and facilitates the repair process [35]. MTA2 appears to be particularly important because DNA damage increases massively when MTA2 is absent [26]. In CIM-MuSC, MTA2 is absent in the nuclei and translocated to the cytoplasm. The mechanism of translocation is unknown, but clearly, MTA2 cannot serve its normal intranuclear function. In the myonuclei expressing γH2AX, MTA2 abutted the DSBs. We also saw the impaired repair of DNA damage after irradiation in CIM-MuSC. This observation is in agreement with the reported role of NuRD in DNA damage [26]. In addition, the lack of histone 1, a prominent finding in CIM-MuSC, leads to an increased number of DSB [36].

When CIM-MuSC were transplanted into immunodeficient mice, a significantly lower number of muscle fibers were generated than with control MuSC. Two possibilities are suitable to explain the difference: First, CIM-MuSC may have an overall diminished potency to create new muscle fibers, or second, a subpopulation of CIM-MuSC escapes the damage during early critical illness and remain healthy. We would argue for the remains of some healthy muscle stem cells in CIM muscle because muscle fibers of regular diameter were built after transplantation, and the satellite cell niche was repopulated with PAX7-positive cells. These are well-recognized features of muscle regeneration [25,37,38,39]. The reason why some MuSC can escape injury would be interesting and relevant for the prevention and therapy of CIM. The impact of satellite cell numbers on long-term recovery after CIM has been described before [6].

The benefit of physical activity is well known. It has been shown that muscle contraction is an epigenetic modifying factor and improves chronic diseases such as metabolic syndrome, diabetes, cancer, cardiovascular, and neurodegenerative diseases [40,41]. The profit of neuromuscular stimulation in the prevention and treatment of critical illness remains a controversial issue [42]. EMS is known to improve glucose metabolism in CIM patients. The method is effective in stimulating muscle protein synthesis and reduces muscle atrophy in CIM patients when performed during the acute phase of the critical illness [9,27]. However, EMS does not have a positive effect on muscle strength when performed after the appearance of ICU-acquired weakness in ICU survivors during rehabilitation [43]. Therefore, EMS appears to only act preventively in the acute phase of critical illness [44]. The MuSC from the stimulated muscle we analyzed was obtained in a small pilot study from the CIM patients who received unilateral EMS [9]. We identified an almost complete absence of epigenetic alterations and DSB when EMS was applied to the muscle. However, the small number of samples does not allow for a solid conclusion. 

Furthermore, the presence or absence of epigenetically altered MuSC in CIM depends on factors intrinsic to the skeletal muscle itself. The fact that the control MuSC came from the unstimulated muscle of the same patient makes the role of circulating factors such as inflammatory cytokines or pharmacological agents unlikely. However, the possibility that MuSC dysfunction, the disintegration of the NuRD complex, and the prevention of DNA damage are all possible opens new optimistic avenues for possible therapeutic interventions in ICU.

## 4. Materials and Methods

### 4.1. Human Muscle Stem Cell Populations

Human muscle stem cells (MuSC) were isolated and expanded from muscle biopsy specimens obtained from the Vastus lateralis [11]. All muscle stem cell populations were > 98% positive for the myogenic marker desmin. Only young muscle stem cell populations with a passage number < 8 were included in the experiments to exclude senescence-related alterations in cultured cells. Research use of the material was approved by the regulatory agencies (EA2/175/17, Charité Universitätsmedizin Berlin; clinical trial number ISRCTN77569430). MuSC were analyzed from previously taken biopsies [9] from 12 CIM patients (thereof *n* = 3 CIM patients underwent unilateral electrical muscle stimulation with a muscle biopsy from stimulated and unstimulated muscle), 3 ICU patients without CIM (ICU control), and 18 healthy individuals with no evidence of neuromuscular disorders or medical history of critical illness (Appendix A). CIM-MuSC and control MuSC were isolated in the same period of time.

The criteria for diagnosis of CIM, as well as the implementation of electrical muscle stimulation (EMS), was described in detail in supplementary material [9].

### 4.2. Satellite Cell Cultures and Techniques

Human satellite cells culture were prepared according to Marg et al. [11]. Briefly, human muscle fiber fragments (HMFF) were isolated from muscle biopsies. After hypothermic treatment (7 days at 5 °C), HMFFs were incubated in a humidified atmosphere containing 5% CO_2_ at 37 °C for up to 3 weeks. Outgrowing activated satellite cells were analyzed between passages 5 and 7. 

**Standard fusion (i.e., differentiation) assay** was performed as described in [11]. In brief, 5000 cells per well were seeded on a 24-well plate and cultured for 2 days in standard muscle cell medium. Afterward, the cells were cultured in either standard muscle cell medium (Skeletal Muscle Growth Medium, Provitro with a final serum concentration of 12.8% after addition of fetal calf serum) or with serum depleted medium (Opti-MEM, Thermo Fisher Scientific, Gibco, serum concentration: 0%) for the next 5 days. The fusion index was determined by dividing the number of nuclei within myotubes by the total number of nuclei counted. For each sample, at least 200 nuclei were analyzed.

For **immunofluorescence** staining, standard protocols were used as described in [11,25] with modifications in incubating time and blocking solution. In brief, the cells were grown on µ-slides (8 well, ibidi) for 2 days, washed with PBS, and fixed with 100% methanol (for Histone 3 trimethyl K9) or 3.7% formaldehyde (for all other antibodies). After permeabilization with 0.2% Triton X-100 for 10 min (for all antibodies except Histone 3 trimethyl K9), the cells were blocked in 1% bovine serum albumin (BSA)/PBS for 1 h and incubated with primary antibodies diluted in 1% BSA/PBS overnight at 4 °C, and subsequent with secondary antibodies diluted in PBS for 1 h at room temperature. Primary and secondary antibodies used for the staining are presented in Appendix A. For the analysis, at least 200 nuclei were counted in each individual primary muscle stem cell culture and analyzed in GraphPad Prism (v7).

**Western blotting** was performed with standard techniques. After protein extraction, proteins were separated with Novex Wedge Well 1 mm Tris-Glycin gel (Thermo Fisher Scientific) and transferred to a nitrocellulose Protran transfer membrane (Carl Roth). The primary antibodies were diluted in 4% milk or 4% BSA in TBS with 0.05% Tween (TTBS) (Appendix A). Fluorescent Western blots were analyzed on a LiCore Odyssey, (Nebraska, USA). For ECL images, immunoblots were incubated with SuperSignal West Dura (Thermo Fisher Scientific) for 1 min and exposed between 10 s and 5 min on a ChemiSmart 5000 (PEQLAB Biotechnologie, Erlangen, Germany).

For **cell proliferation analysis**, MuSC were pulsed with 2 mM carboxyfluorescein succinimidyl ester (CFSE) for 10 min at 37 °C and then seeded at a concentration of 3 × 10^4^/^mL^. Cells were analyzed by FACS. We performed an **γH2AX foci assay**. MuSC was grown on cover slips overnight. Cells were irradiated (5 Gy IR) and allowed to recover for the indicated times. Upon fixation with 4% paraformaldehyde and permeabilization with 0.5% Triton X-100, cells were stained with anti-γH2AX antibody, with goat anti-rabbit Alexa 546 and goat anti-mouse Alexa 488 as secondary antibodies. DNA was counterstained with Pentahydrate (bis-Benzimide) (Hoechst). Images were acquired using inverted LSM700 laser scanning confocal microscope (Zeiss). 

**RNA isolation and quantitative PCR** were performed with standard techniques. RNA isolation and analysis were prepared according to the protocol published in [25]. Primer sequences are listed in Appendix A. GAPDH, cyclophilin A, and L13a were used as reference genes with maximum difference of 10% for all normalized calculations. Generation of the RNA Sequencing libraries and RNA-Seq analysis was carried out as described in Marg et al. [25].

### 4.3. In Vivo Transplantation

Five-week- to seven-week-old male NOD.Cg-*Prkdc^scid^Il2rg^tm1Sug^*/JicTac (NOG) mice were purchased from Taconic Biosciences one week before experiment (*n* = 3 mice for transplantation of CIM-MuSCs, *n* = 4 for Control-MuSCs). Irradiation and cell transplantation were performed using our established protocol [25] (brief description in Appendix A [45,46,47]). All animal experiments were performed under the license number G0035/14 (LAGeSo, Berlin, Germany) [25].

### 4.4. ATAC-Seq

Analysis of the ATAC-Seq dataset was done using the PiGx-ChIP-seq pipeline (version 0.0.20), which performs quality control, preprocessing, mapping, and peak calling for ChIP and ATAC-Seq analysis. Reads were adapter trimmed using Trim Galore (version 0.4.5) and then aligned against the canonical hg19 version of the human genome with omitted mitochondrial chromosome using bowtie2 in local mode with maximum fragment length set to 2000 bp, disallowance of discordant alignments and search restricted to paired-end alignment (version 2.3.2, ’-X 2000 -local -no-discordant -no-mixed’). Mapped reads were filtered for mapping quality above 10 and then duplicated reads were removed using samtools software (version 1.8) (Appendix A). Peaks were called for each individual sample using MACS2, where we followed the authors’ suggested parameters to find enriched cutting sites or, in the case of ATAC-Seq data nucleosome-free regions (version 2.1.0.20151222, --keep-dup auto -q 0.05 --format BAM --nomodel --shift -100 --extsize 200 --gsize hs). In addition, peaks for pooled samples of each group were called using a relaxed significance threshold of 0.1 (the same command as described above, but with -q 0.1) (Appendix A). In order to generate a set of reliable peak regions for further analysis, a so-called “group specific consensus set” for each of our different groups was derived. This was done by overlapping and merging the different sets of sample-specific peaks using the findFeatureComb function of the Bioconductor package genomation (version 1.12.0) and keeping only those peaks that have been detected in all samples of a group. Then we removed any of the consensus peaks overlapping blacklisted regions (version 1) for the human hg19 genome generated as part of the ENCODE Project as these regions are often found at specific types of repeats such as centromeres, telomeres, satellite repeats, and thus tend to show an artificially high signal for short-read sequencing. This left us with a set of 74,292 distinct peaks in total, which are covered by any of the “group specific consensus sets” that we took as the “final/combined consensus set” and used for any downstream analysis that includes a consensus set (Appendix A). Additional information is described in Supplement.

### 4.5. Statistical Analysis

For statistical calculations and plots, we used GraphPad Prism (v7). Group differences were tested using a Mann–Whitney U test for independent distributed samples. To evaluate the correlation of categorical variables, we used a Chi-square test. This test was applied to test the influence of electrical muscle stimulation on the presence of Ki-67, PAX7, H3K9me3, γH2AX, MTA2, RBBP4 in the myonuclei of control- and CIM-MuSC. All other group differences were tested using a Mann–Whitney U test for independently distributed samples. Test results with a *p*-value <0.05 were considered statistically significant. The analysis of ATAC-Seq data is presented in detail separately (Supplement, chapter ATAC-Seq). RNA-Seq and ATAC-Seq data were uploaded to the server of the European Genome-Phenome Archive. 

## 5. Conclusions

Our results demonstrate global and lasting epigenetic changes in human muscle stem cells that occur during the first days of critical illness, therefore providing a valid explanation for the insufficient capacity to regenerate muscle strength in CIM patients. The discovery of severe epigenetic alterations in skeletal muscle stem cells as an explanation of poor recovery after CIM opens new avenues for therapeutic interventions. 

## Figures and Tables

**Figure 1 ijms-24-02772-f001:**
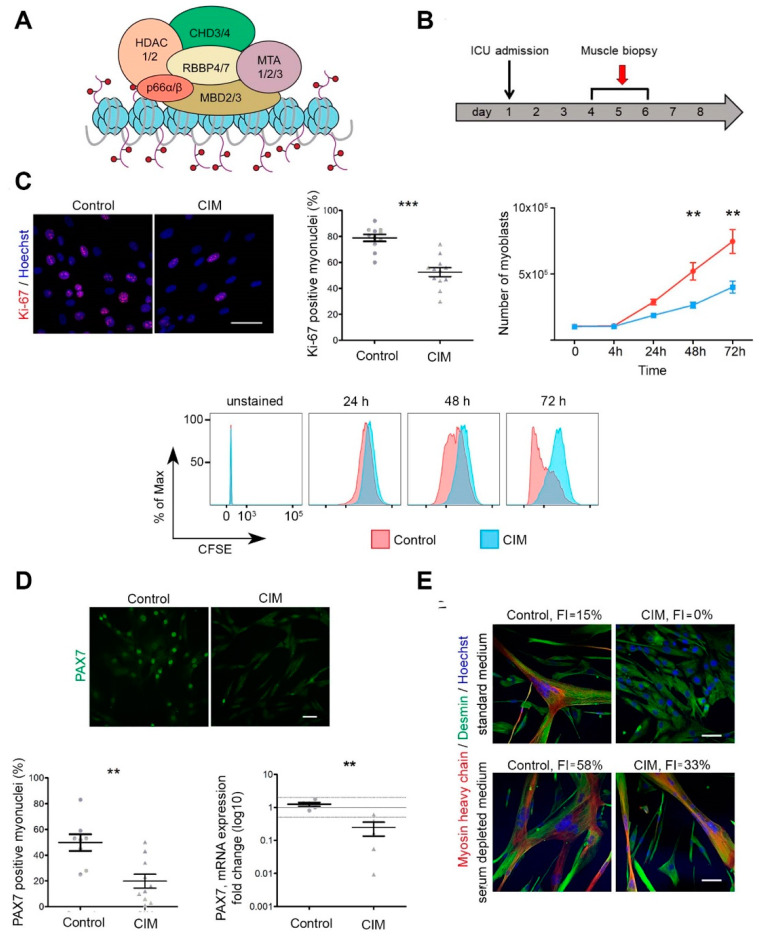
Impaired proliferative and differentiating capacity of CIM-MuSC. (**A**) Schematic overview of NuRD complex composition in muscle tissue. (**B**) Study design. MuSC originated from ICU patients (CIM, *n* = 12; ICU controls without CIM, *n* = 3). (**C**) Left: Immunofluorescent staining for Ki-67 (red), Hoechst (blue), and quantification (CIM, *n* = 12, mean 52%; control *n* = 11, mean 79%; *p* = 0.0001). Right and bottom: Carboxyfluorescein succinimidyl ester (CFSE) cell proliferation assay in CIM- (blue) and control MuSC (red) analyzed by FACS at 24 h, 48 h, 72 h; (*n* = 4 for CIM- and control MuSC; 24 h: control 2.7 × 10^5^, CIM 2.2 × 10^5^; 48 h: control 5.2 × 10^5^, CIM 2.6 × 10^5^, *p* < 0.01; 72 h: control 7.5 × 10^5^, CIM 4.1 × 10^5^, *p* < 0.01). (**D**) Decreased expression of transcription factor PAX7 in CIM-MuSC. Immunofluorescent staining for PAX7 (green) (CIM, *n* = 10, mean 20%; control, *n* = 8, mean 50%; *p* < 0.01); quantitative PCR of PAX7 (CIM, *n* = 5; control, *n* = 5; *p* < 0.01). (**E**) Reduced fusion capacity of CIM-MuSC after 7 days in standard muscle cell medium (*p* < 0.001) or after 5 days in serum-depleted medium (*p* < 0.001); *n* = 4 for CIM- and control MuSC. Immunofluorescent staining for desmin (green), myosin heavy chain (red), and Hoechst (blue). FI—Fusion index. ** *p* < 0.01. *** *p* < 0.001.

**Figure 2 ijms-24-02772-f002:**
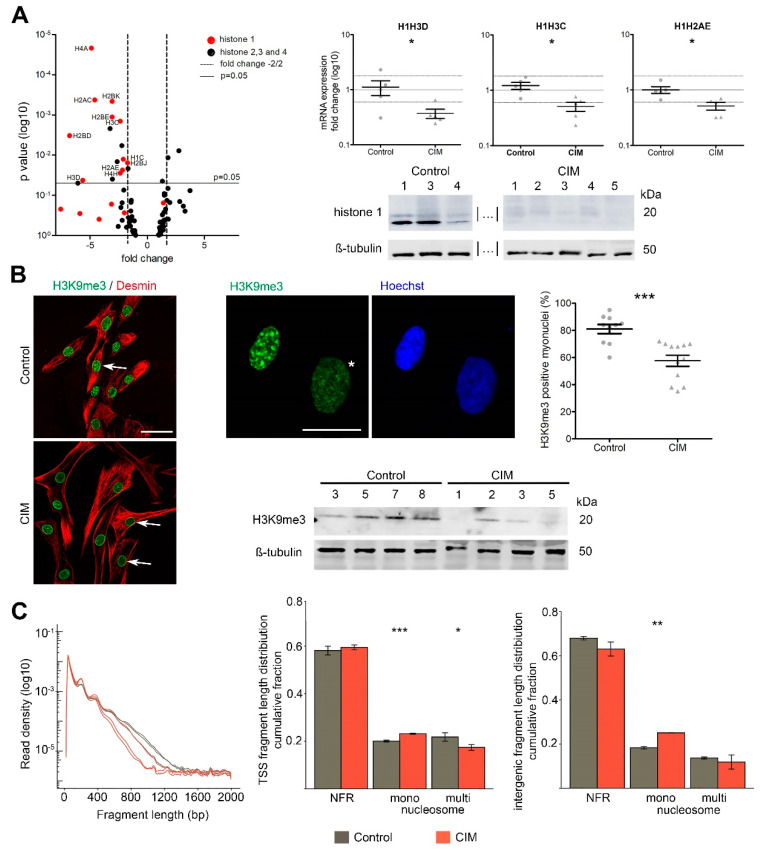
Opened chromatin state in CIM-MuSC. (**A**) RNA-Seq analysis of CIM and control MuSC. Confirmation by qPCR and on protein level by Western blot. Additional samples on the blot indicated by │…│ are shown in Figure 5. (**B**) Immunofluorescent staining for desmin (red), H3K9me3 (green), Hoechst (blue); bar 50 µm. Arrows: control: myonucleus positive for H3K9me3; CIM—absence of H3K9me3. Zoomed images: star: myonucleus from control MuSC with euchromatin, scale bar 10 µm. On the right quantification of myonuclei positive for H3K9me3 (CIM, *n* = 12, mean 57%; controls, *n* = 10, mean 81%; *p* < 0.001). Western blot for H3K9me3. (**C**) ATAC-Seq analysis of CIM- and control MuSC. Global fragment length distribution plotted against read density to locate the expected open chromatin region (peak at < 100 bp) followed by nucleosome phasing pattern (peak every ~200 bp). Intergenic. Mono, single nucleosome spanning; multi, multiple nucleosome spanning; TSS—Transcription Start Site; NFR—Nucleosome Free Region. Data information: All values are expressed as mean ± SEM. In (**B**), at least 200 nuclei/cell population were counted for quantification. In (**A**,**B**), data were analyzed by Mann–Whitney U test; in (**C**), by t-test. * *p* < 0.05, ** *p* < 0.01, *** *p* < 0.001.

**Figure 3 ijms-24-02772-f003:**
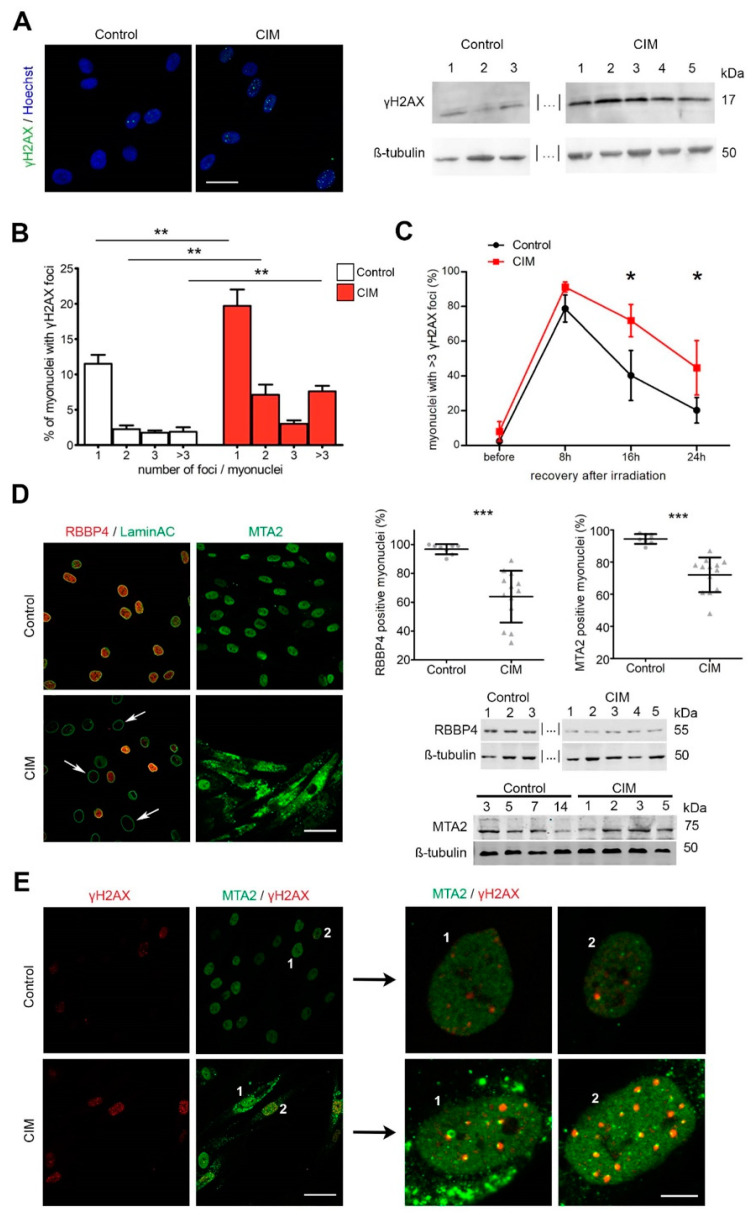
DNA-double strand breaks (DSBs) and impaired nucleosome remodeling and deacetylase (NuRD)-complex in CIM-MuSC. (**A**) Immunofluorescent staining for γH2AX (green) and Western blot analysis of γH2AX in CIM-MuSC and controls. Two other samples, indicated by |… |, are shown in Figure 5. (**B**) Quantification of myonuclei with DSBs (γH2AX foci >3) in CIM-MuSC (*n* = 10) and controls (*n* = 8). (**C**) Analysis of γH2AX foci accumulation kinetics shows reduced repair of irradiation-induced DSBs in CIM-MuSC (*n* = 4 for CIM- and control MuSC; 8 h: CIM 91%, control 78%; 16 h: CIM 72%, control 40%, *p* < 0.05; 24 h: CIM 40%, control 20%, *p* < 0.05). (**D**) Immunofluorescent staining for NuRD components RBBP4 (red) (CIM, *n* = 13, mean 64%; control, *n* = 8, mean 97%; *p* < 0.001) and MTA2 (green) (CIM, *n* = 13, mean 72%; controls, *n* = 8, mean 94%; *p* < 0.001) and Western blot analysis. Arrows: RBBP4 negative myonuclei. (**E**) Co-localization of γH2AX (red) and MTA2 (green) by immunofluorescence in myonuclei. MTA2 accumulated close to DSB The right panel shows a magnification of the cells marked with 1 and 2 on the left panel. Data information: In (**A**,**D**), scale bar for all images: 50 µm. In (**E**) scale bar of the images on the left: 50 µm; scale bar of zoomed images on the right: 5 µm. All values are expressed as mean ± SEM. In (**B**,**D**), at least 200 nuclei/cell population were counted for quantification; data were analyzed by Mann–Whitney U test. In (**C**), data were analyzed using paired t-test. * *p* < 0.05, ** *p* < 0.01, *** *p* < 0.001.

**Figure 4 ijms-24-02772-f004:**
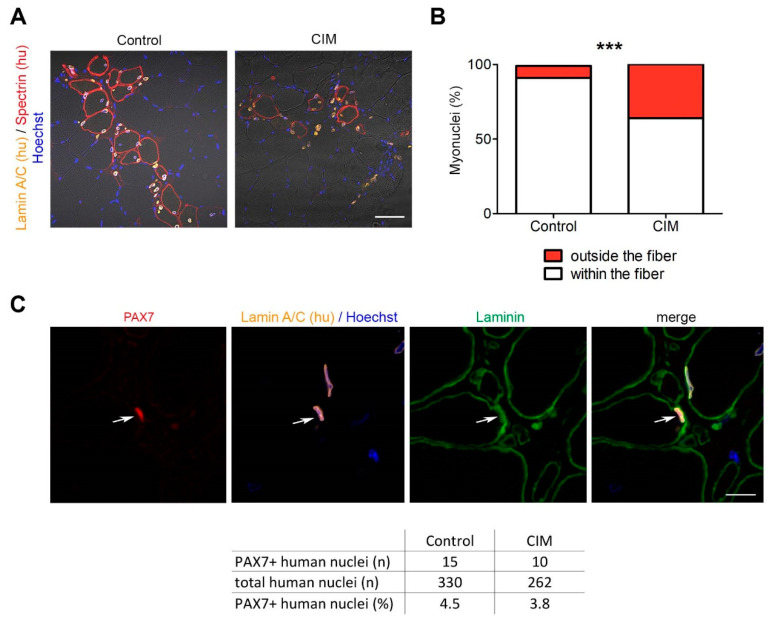
Reduced regenerative capacity of CIM-MuSC in vivo. (**A**) CIM-MuSC (1 × 10^5^) was transplanted into the TA muscle of NOD mice. Immunofluorescent staining three weeks after transplantation: human Lamin A/C (yellow), human spectrin (red), Hoechst (blue) (CIM, *n* = 3 mice; control, *n* = 4 mice). Scale bar 50 µm. (**B**) Quantification of human nuclei integrated into muscle fibers (white) and cells that failed to integrate into muscle fibers (red). Chi-square test; *** *p* < 0.0001; >300 nuclei were counted per condition. (**C**) PAX7 positive human myonuclei located in their anatomical niche (red: PAX7, blue: Hoechst, green: Laminin, orange: human Lamin A/C). Arrow shows the same nuclei. Scale bar 10 µm. Quantification of PAX7-positive cells within the stem cell niche.

**Figure 5 ijms-24-02772-f005:**
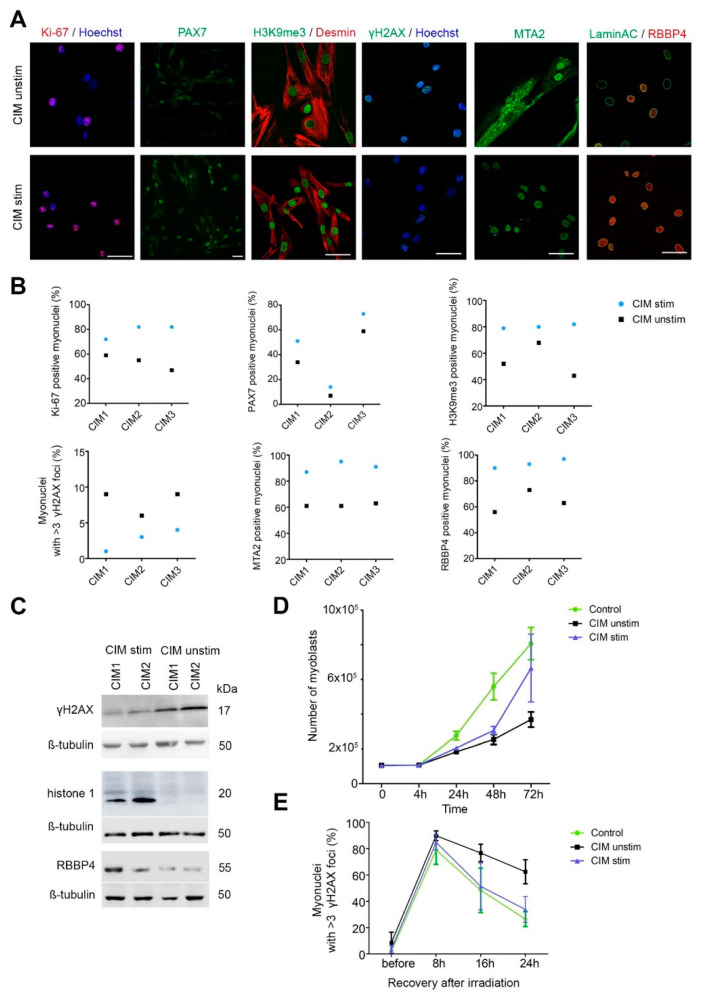
Effect of electrical muscle stimulation (EMS) on CIM-MuSC. Analyses were performed on EMS-stimulated MuSC (CIM stim) and nonstimulated contralateral controls with CIM (CIM unstim). (**A**) Immunofluorescence staining of Ki-67, PAX7, H3K9me3, γH2AX, MTA2 and RBBP4. (**B**) Quantification of myonuclei positive for Ki-67, PAX7, H3K9me3, MTA2, RBBP4 and myonuclei with >3 γH2AX foci. Electrical stimulation has a significant influence on these parameters. (**C**) Western blot analysis of γH2AX, histone 1, and RBBP4. (**D**) Proliferation analysis of Carboxyfluorescein succinimidyl ester (CFSE) cell proliferation assay in CIM-MuSC from stimulated muscle (*blue*), CIM-MuSC from unstimulated muscle (*black*), and control MuSC (*green*) analyzed by FACS at 24 h, 48 h, 72 h (*n* = 3 for CIM- and control MuSC). (**E**) Analysis of γH2AX foci accumulation kinetics shows the differences in repair of irradiation-induced DSBs in CIM- and control MuSC (*n* = 3 for CIM- and control MuSC). Data information: Scale bar for all images: 50 µm. In (**B**), for all, analysis was performed on at least 200 nuclei/cell colonies. In (**D**,**E**), data are presented as mean ± SEM (the following categories were compared: CIM stimulated; CIM unstimulated).

**Table 1 ijms-24-02772-t001:** Donor Characteristics.

Patient/MuSC Donor	Sex/Age (Years)	CIM	Diagnosis	Electrical Stimulation
CIM1	M, 67	yes	ARDS	yes
CIM2	F, 48	yes	Sepsis	yes
CIM3	M, 64	yes	ARDS	yes
CIM4	M, 74	yes	ARDS	no
CIM5	M, 53	yes	ARDS	no
CIM6	M, 67	yes	Sepsis	no
CIM7	M, 41	yes	ARDS *	no
CIM8	M, 36	yes	ARDS *	no
CIM9	M, 54	yes	Trauma	no
CIM10	M, 41	yes	ARDS	no
CIM11	M, 42	yes	ARDS	no
CIM12	M, 67	yes	Sepsis	no
ICU1	M, 18	no	Trauma	no
ICU2	M, 50	no	Intracranial bleeding	no
ICU3	W, 69	no	ARDS	no

ARDS—acute respiratory distress syndrome. * H1N1 influenza. All patients required anti-inflammatory therapy upon arrival to the intensive care unit.

**Table 2 ijms-24-02772-t002:** Characterization of patients within groups: ICU patients with and without CIM and controls.

	CIM *n* = 12	ICU Controls *n* = 3	Controls *n* = 18
Age (years): mean(range) median SD	54.5 (36–74) 53.5 13	45.7 (18–69) 45.7	45.8 (18–66) 48.5 12.3
Gender: male/female, no. (%)	11(91.6)/1(8.4)	2 (66.6)/1(33.3)	14(77.8)/4(22.2)
SOFA score, ICU admission: mean (range)	11 (7–19)	14 (9–17)	n/a
SAPS II score, ICU admission: mean (range)	57.8 (24–98)	55 (40–75)	n/a
Length of ICU stay (days): mean (range)	42.8 (5–131)	31.7 (25–34)	n/a

SOFA—sequential organ failure assessment; SAPS-II—simplified acute physiology score-II; n/a—not available.

**Table 3 ijms-24-02772-t003:** Results of RNA-Seq analysis for three different gene clusters.

Histone Genes			Heat Shock Associated Genes			Immune Response Genes		
Name	Fold Change	*p*-Value	Name	Fold Change	*p*-Value	Name	Fold Change	*p*-Value
HIST1H4A	4.880	0.000	MMP3	65.673	0.000	IL1B	81.724	0.000
HIST1H2AC	4.588	0.000	ATP2A3	5.054	0.000	MMP12	136.679	0.000
HIST1H2BK	3.072	0.000	GLRX	2.841	0.000	ITGAX	35.469	0.000
HIST1H2BE	3.072	0.001	KDR	90.280	0.000	IL8	14.265	0.000
HIST1H3C	2.357	0.001	IL1B	88.971	0.000	CTSK	9.848	0.000
HIST3H2A	3.251	0.002	SLC16A6	28.814	0.000			
HIST1H2BD	6.783	0.003	ITGAX	38.387	0.000			
HIST2H2BE	2.221	0.006	EPAS1	3.495	0.000			
HIST1H1E	−2.771	0.008	DUSP6	2.448	0.000			
H2AFX	−1.818	0.012	TIMP1	2.686	0.000			
HIST1H1C	2.098	0.013	CASP4	2.333	0.000			
HIST2H4B	2.625	0.015	FTL	3.006	0.000			
HIST1H2BJ	1.718	0.016	DMKN	7.315	0.000			
H2AFJ	1.672	0.022	PAPPA	11.881	0.000			
HIST1H2AE	2.164	0.024	BCL2A1	42.596	0.000			
HIST1H4H	2.360	0.028	CTBS	2.278	0.000			
HIST4H4	3.042	0.040	TLR2	7.233	0.000			
HIST1H3D	5.644	0.042	CPM	3.483	0.000			
H2AFZ	−1.330	0.045						
H3F3C	6.065	0.050						

Fold changes show differences between group CIM-MuSC versus control (=1). A negative fold change indicates that this gene is downregulated within the CIM-MuSC group compared to the control group, whereas a positive sign means upregulation for that gene within the CIM-MuSC group. Genes are defined as up- or downregulated if absolute numbers are above 2.

## Data Availability

The data that support the findings of this study are available on request from the corresponding author [J.S. and S.S.], upon reasonable request. The data are not publicly available due to restrictions (because they contain information that could compromise the privacy of research participants).

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
