# Peer review of "Disintegration of the NuRD Complex in Primary Human Muscle Stem Cells in Critical Illness Myopathy"

_ijms, 2023, doi:10.3390/ijms24032772_

Round 1

Reviewer 1 Report

In this work, Schneider and co-authors proposed a possible mechanism for CIM in terms of epigenetic alterations in MuSCs. They isolated MuSCs from early CIM and analyzed the epigenetic alterations with various techniques. Their results showed that the CIM-MuSCs possessed Impaired growth and differentiation, lost heterochromatin, accumulated double-strand breaks in DNA, and altered NuRD subunits. They further investigated the muscle regenerative potential by transplanting MuSCs into immunodeficient NOG mice, and found that CIM-MuSC exhibited significant growth deficits, reduced differentiation impaired DNA repair, and failed to build new muscle fibers. In general, the topic of this work is meaningful, and the methodology for studies in vitro and in vivo were acceptable, but there were concerns that should be addressed, mainly in the sampling procedure:

1.      CIM is an acquired illness during ICU stay. How could the patients be diagnosed as CIM with only 4-6 day after ICU admission? What were the Indications for diagnosis? The authors did not provide enough information in detail.

2.      From Table 1, the number of valid CIM samples should be 9. There were different claims in various locations in this manuscript. In Abstract, it was claimed n=15; in the Results Part, n=12. Please check carefully and make sure the correct description was used. In addition, from those 3 electrically stimulated samples, it was not able to declare that “MuSC obtained from electrically stimulated muscle of CIM patients were undistinguishable from control MuSC”.

3.      The patients were mainly male with only on female (in the electrical stimulation group). I think this is not proper. If the authors had some special consideration, please describe it in the text.

4.      Ki-67 is generally evaluated in determining the malignancy of tumor. Was it suitable here for evaluating the proliferation of stem cells? I suggest to check the growth using MTS ot CCK8 kit that are usually used for evaluate cell proliferation.

5.      Although the results of in-vitro studies were in relevant with epigenetic alterations, there was, actually, no enough data to support the epigenetic alterations of CIM-MuSCs. It would be better to use other description.

Author Response

We greatly appreciate your comments and would like to thank you for your time, constructive criticism and fast processing.

Please see the attachment for point-by-point response.

Reviewer 2 Report

In the submitted manuscript the authors report about extensive investigations to test the Hypothesis that epigenetic mechanisms could contribute to the failure of skeletal muscle regeneration in critical illness myoptahy patients.

The scientific background is well described, the hypothesis is clear and valid and the methods used to test the hypothesis are adequate. For all findings  several methods were used to strenghten and confirm the obtained results in order to be able to make a solid conclusion. The results are well documented and comprehensible.

I only have a few very minor remarks:

Title:

Would change the title to "Desintegration of the NuRD complex in primary human muscle stem cells in crital illness myopathy"

Results section

Line 130: Did you reall mean to say " in the absence of serum depleted medium" or should it mean "in serum depleted medium"?

Methods section:

Please add "in brief" descriptions of the methods for Satellite cell culture, Standard fusion assay and immunofluorescence.

Immunofluorescence: Please provide citation of standard protocol used or refere to user manual.

Line 463: Did you mean to say "7 five-week old male..."

 Discussion section

Line 371 should it be features instead of feature?

Author Response

(The authors gave the same response as above.)

Round 2

Reviewer 1 Report

The authors had revised the manuscript and dissolved most of my previous concerns. I think it could be accepted for publication after minor modification.  Specifically, although histone 1 downregulation is closely related with epigenetic alterations, there are actually other possibilities. Therefore, I suggest to carefully discuss the mechanism associated with epigenetic alterations.

Author Response

We added a paragraph to the introduction about the different functions of Histone 1 (page 5) and introduced two further references. The introduction complements now the sections on Histone 1 in Results (page 7 and 8) and Discussion (page 20).